# Initializing and Retrofitting Key-Value Adaptors for Traceable Model Editing

## Abstract

As the insight of knowledge storage in language models deepens, the ability to perform CRUD (Create, Read, Update, Delete) operations on language models becomes increasingly indispensable for satisfying the demands of managing rapidly updating knowledge. Considering the high cost of fine-tuning language models, model editing methods with low cost are usually required to manipulate models' knowledge. Evident suggests that modules carrying knowledge in a Transformer module are primarily the MLP blocks, thus we propose **iReVa**, a method that explicitly initializes and retrofits key-value pairs into MLP blocks to construct a new mapping of a piece of knowledge without damaging the irrelevant knowledge. In comparison to existing methods, iReVa reveals better interpretability and stronger capacity for carrying traceable edits. Experiment results on series of GPT series models show our prominent performance on edit success and generalization without influencing specificity. We also perform the first attempt at conducting knowledge withdrawal test of iReVa. Our codes are available at github.com/timberflow/iReVa.git.

## 1 Introduction

Language Models (LMs) [1] are becoming imperative tools for consulting in real-world scenarios. One significant reason for the prevalence of LMs is their ability to answer factoid questions. For example, when we ask an LM with the question "*Who is president of America ?*", it returns the answer "*Joe Biden*". Even though a mass amount of knowledge is stored in the LMs, we still face the issue of out-of-date and missing knowledge [2, 3]. Alternatively, some knowledge may change over years and some domain-specific knowledge may be absent from the LMs.

To bridge the gap, the task of model editing is introduced to *edit* the knowledge in LMs, which targets at conducting change to the parameters of LMs and inject certain knowledge to them [4]. The difficulty of this task lies in the manipulation to the LMs, where the knowledge is implicitly stored in dense vectors. A naive solution to model editing is fine-tuning a LM with the new knowledge, whereas the cost is climbing with the surging size of LMs. More recent studies propose to directly update the models' weights in mastery phase [5, 6] via either teaching a hyper-network to learn the change of the weights or locating-then-editing knowledge neurons [7, 8, 9, 10]. While the editing methods above are efficient in updating knowledge in LMs, they encounter the difficulties of differentiating the existing and new knowledge, which makes the editing hard to control. Methods like life-long model editing [11], MELO [12], and T-Patcher [13] propose to learn the representation for new knowledge and merge this information with the original models.

However, these methods still conform to the paradigm of learning the batch edit [13, 14] as a whole without modeling edit parameters in a traceable way, which can not conform the edit success to each edit and have a lack interpretability to the editing. In contrast, we propose a method of

**I**nitializing and **R**etrofitting KE**y**-**V**alue **A**daptors (**iReVa**), an editing method that inserts a key-value adaptor to indicate the mapping of an edit data pair and further retrofit the adaptor with multiple objectives. Moreover, to prevent the unnecessary change to the irrelevant knowledge, we elaborately design activation mechanism for the knowledge neurons. Experimental results on series of GPT-like models show that iReVa is able to outperform the SOTA results by around 9% and 6% average score improvement on zsRE-10K and PARAREL-10K, respectively. Moreover, iReVa is able to perform knowledge withdrawal in almost perfect condition.

Our contributions are summarized as follows: 1) We introduce a novel editing method which initializes and retrofits a key-value adaptor for traceable model editing, which is compatible to most LMs. 2) Our method outperforms recent baselines on model editing tasks with noticeable margins based on various evaluation metrics. 3) We validate the interpretability and generalization capabilities of our method by conducting further analysis such as knowledge withdrawal test and generalization test.

## 2   Related Work

### 2.1   Insight of Knowledge Storage in Language Models

As pre-trained LMs show strong abilities to answer factoid questions. Discussion about how LMs store knowledge has emerged. [2] introduced the perspective of treating LMs as knowledge bases and proved its plausibility, which attracted the subsequent attention towards the exploration on the form of knowledge incorporated by LMs. The opinion pointed out by [15] indicates that factual knowledge is stored in two-layer-FFN network of a Transformer due to the similar form as key-value memories. This opinion was followed by [16], which further derives the coefficient between final prediction and knowledge neurons in MLP blocks. In contrast, [9], through a casual-tracing experiment, posed viewpoints that knowledge is stored in self-attention module. [7] further validates that the weight update is concentrated on parameters in self-attention module when we train models with new knowledge. Our editing method is built upon the former hypothesis and we focus on the editing to the MLP blocks.

### 2.2   Editing LMs by Manipulating Knowledge

With the frequent update of the knowledge, the demand of model editing increases. Diverse studies have been proposed. By analogy with human knowledge acquisition, we can categorize the editing into three distinct phases. In recognition phase [17], methods such as ERAC and IKE [8, 18] solved the problem by importing additional memories in the form of a relevant contexts or prompts. In association phase [6], parameter-efficient tuning [19, 20, 12, 11] inserts low-rank adaptors or prefix token embeddings to fine-tune new knowledge and combine them to the original models. There are also some studies directly changing the weights of Transformers in mastery phase [5]. For example, [7] proposed KE and [8] proposed MEND to predict the updated parameters of a model with a trained hyper-network. Furthermore, ROME [9] and MEMIT [10] compute the weight update explicitly with proper representations of knowledge queries and values. However, none of them focuses on traceable model editing, which allows more flexible manipulation of the knowledge.

## 3   Problem Formulation

We follow the previous studies [21, 12, 11] to formulate the task. Suppose we are given a base model that could be a pre-trained language model $f_\Phi$ parameterized by $\Phi$, model editing aims at editing $f_\Phi$ with a dataset $\mathcal{D}_{in} = \{(x_1, y_1), ..., (x_i, y_i)..., (x_n, y_n)\}$, where $(x_i, y_i)$ denotes the edit input-output pairs. Initially, for $x_i \in \mathcal{D}_{in}$, the base model makes prediction $\hat{y}_i = f(x_i)$ but $\hat{y}_i \neq y_i$. In this case, we change $f_\Phi$ by *editing* its parameters to $\Phi^*$. A good model editing to $f_{\Phi^*}$ should satisfy: 1) for any $x_i \in \mathcal{D}_{in}$, the edited model $f_{\Phi^*}$ should output desired predictions, that is $f_{\Phi^*}(x_i) = y_i$; 2) for any input out of the scope of $\mathcal{D}_{in}$, which is denoted as $\mathcal{D}_{out}$, the edited model $f_{\Phi^*}$ should retain the original predictions, that is $f_{\Phi^*}(x_i) = f_\Phi(x_i)$; 3) the edit of $(x_i, y_i)$ towards $f_{\Phi^*}$ should not influence any prior edits $x_{<i} \in \mathcal{D}_{in}$.

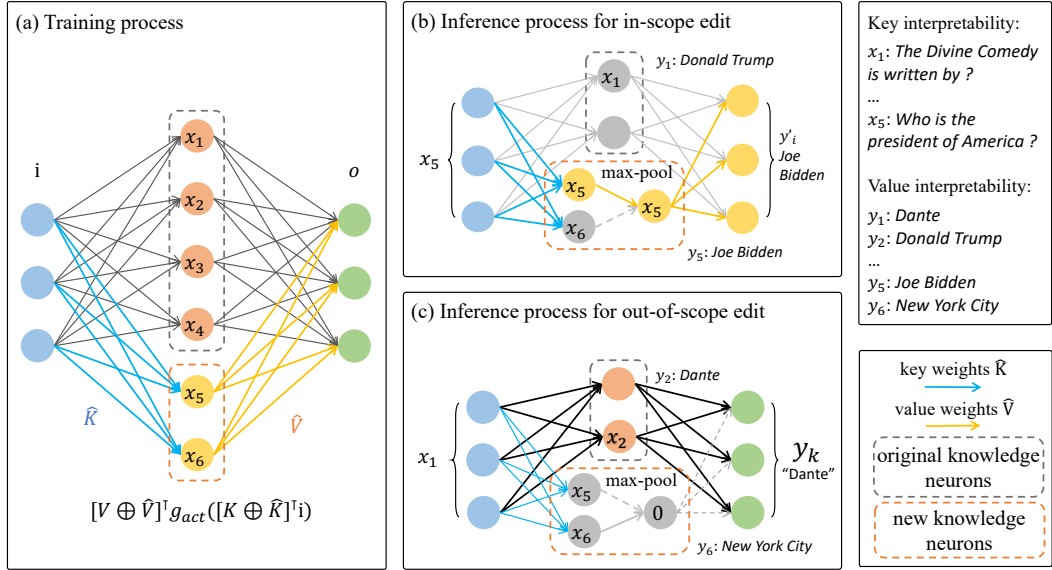

Figure 1: Architecture of iReVa. The left block shows the training procedure with the newly inserted knowledge neurons. The middle block shows the inference procedure with in-scope and out-of-scope edits. We interpret the inference phase by giving some explicit examples (Please note we omit some neurons during inference due to the space limit.). When the query falls in the in-scope edit, our key-value adaptor will be activated and retrieve the corresponding knowledge. When the query falls in the out-of-scope edit, our key-value adaptor is inactive and the model retrieves knowledge from the original memory.

## 4 Method

To develop an editing method that supports traceable edits to knowledge neurons, we introduce a novel method "**iReVa**" that **i**nitializes and **Re**trofits k**E**y-**V**alue **A**daptors for traceable model editing. The pre-trained LM $f_\Phi$ usually contains Transformer blocks, which consist of intertwined self-attention and feed-forward layers. The prior studies [15] have shown that the inside MLP blocks are commonly deemed as the neurons for storing implicit knowledge. Our method is able to insert new knowledge but without damaging the irrelevant knowledge in the models by inserting and retrofitting the key-value adaptors to these blocks.

Figure 3 depicts the architecture of our proposed method. For a two-layer-FFN MLP block in the $l$-th layer of the original model $f_\Phi$, we denote the weights of the first FFN layer as $\mathbf{K}^l \in \mathbb{R}^{d_1 \times d_2}$ and the second FFN as $\mathbf{V}^l \in \mathbb{R}^{d_2 \times d_1}$. Assume a hidden state $\mathbf{h}^l \in \mathbb{R}^{d_1}$ is an input of the FFN of $l$-th layer, the above block processes the input as follows:

$$\mathbf{i}^l = \text{LAYER\_NORM}(\mathbf{h}^l + \text{SELF\_ATTN}(\mathbf{h}^l)) \tag{1}$$

$$\mathbf{o}^l = \mathbf{V}^{l\intercal} g_{act}(\mathbf{K}^{l\intercal}\mathbf{i}^l) \tag{2}$$

$$\mathbf{h}^{l+1} = \text{SELF\_ATTN}(\mathbf{i}^l + \mathbf{o}^l) \tag{3}$$

where $g_{act}$ is the activation layer and $\mathbf{h}^{l+1} \in \mathbb{R}^{d_1}$ is the input of the next Transformer block. Here, $\mathbf{K}^l$ and $\mathbf{V}^l$ emulate neural memories, where keys capture input patterns and values are stored knowledge to be retrieved. When there comes an input vector, it first computes a distribution over the keys, then retrieve the expected knowledge. As the process is just the same for each layer, we can choose any of the layers to edit, we omit $l$ for simplicity in the following description.

Our method inserts a key-value adaptor into the existing MLP block. Specifically, we update $\Phi$ by inserting a new knowledge neuron to store the edit. Two matrices $\hat{\mathbf{K}} \in \mathbb{R}^{d_1 \times n}$ and $\hat{\mathbf{V}} \in \mathbb{R}^{n \times d_1}$ perform as the key-value pair to memorize $n$ edited knowledge, where the knowledge is well-indexed by $n$ dimensions. Therefore, Equation 2 becomes:

$$\mathbf{o} = [\mathbf{V} \oplus \hat{\mathbf{V}}]^\intercal g_{act}([\mathbf{K} \oplus \hat{\mathbf{K}}]^\intercal \mathbf{i}) \tag{4}$$

$$= \mathbf{V}^\intercal g_{act}(\mathbf{K}^\intercal \mathbf{i}) + \hat{\mathbf{V}}^\intercal g_{act}(\hat{\mathbf{K}}^\intercal \mathbf{i}), \tag{5}$$

where $\oplus$ denotes concatenation. As we can see, the key-value adaptor appends more information to
**o**, which could overwrite the original output. And original parameter set $\Phi$ is extended to $\Phi^*$ with the
new included parameters $\hat{\mathbf{K}}$ and $\hat{\mathbf{V}}$. Therefore, we aim to find a good key-value adaptor for model
editing that can collaborate with the original knowledge neurons. Considering the independence of
the above two function terms and the potential more flexible combination to the output, we relax
the formulation of the adaptor to $\text{ADAPTOR}(\mathbf{i}; \hat{\mathbf{K}}, \hat{\mathbf{V}}) = \alpha \hat{\mathbf{V}}^\intercal g_{act}(\hat{\mathbf{K}}^\intercal \mathbf{i})$, which may be a more
expressive function with a scaling factor $\alpha$ [19]. Next, we will introduce how to find such an optimal
adaptor which not only satisfies the edit success but also preserves the original model behavior.

## 4.1   Initial Key-Value Adaptors for In-Scope Editing

Given an edit $(x_i, y_i) \in \mathcal{D}_{in}$, we first initialize its knowledge neuron $\hat{\mathbf{k}}^0 \in \mathbb{R}^{d_1}$ and $\hat{\mathbf{v}}^0 \in \mathbb{R}^{d_1}$. For
$\hat{\mathbf{k}}^0$, we initialize each key to the $x_i$ using the cached input $\mathbf{i}$ predicted by $f_\Phi(x_i)$ at layer $l$, which
results in a high probability of matching to the input pattern. For $\hat{\mathbf{v}}^0$, we initialize it using the weights
corresponding to $y_i$ from the last layer of $f_\Phi$. Specifically, $f_\Phi(x_i)$ takes charge of generating the next
token which can be deemed as the prediction to $x_i$. Thus, we extract the corresponding column of
the ground truth token $y_i$ from the weights $\mathbf{W} \in \mathbb{R}^{d_1 \times |V|}$ for generating the next token distribution,
where $|V|$ and $d_1$ are the sizes of the vocabulary and dimension of the last layer, respectively [1] After
initialization, we build a mapping from $x_i$ to $y_i$ in a Transformer.

## 4.2   Retrofit Adaptors for Model Editing (Training Phase)

To prevent the effect of the inconsistent scaling brought by built-in parameters in Equation 1, we first
normalize $\mathbf{i}$ to ensure that its mean value is close to 0 before it is fed into the adaptor. Given $(x_i, y_i)$,
we can have the initialized key-value adaptor as follows:

$$\text{ADAPTOR}(\mathbf{i}; \hat{\mathbf{K}}, \hat{\mathbf{V}}) = \alpha (\hat{\mathbf{v}}^0)^\intercal g_{act}((\hat{\mathbf{k}}^0)^\intercal \mathbf{i}).$$

To avoid the inserted key-value adaptor distracts the original knowledge stored in the existing neurons,
we propose to use the activation functions that can activate the memory with a large matching
value and ignore the memory with a small matching value. When we deploy the adaptor to models,
the activation function usually remains consistent with the base model. Furthermore, we apply a
hyper-parameter margin $\theta > 0$, which allows memory to be active if $x > \theta$, otherwise inactivate. For
example, we use GeLU [22] for GPT [23] series model and our activation function can be denoted as:

$$g_{act}(x) = \text{GeLU}(x - \theta). \tag{6}$$

The motivations behind the above design in our activation function are two-fold: First, the activation
function works as a neuronal inhibitor to inhibit the activation of new knowledge neurons, which
retains the original output in most cases. Second, the involvement of the margin further raises the bar
to activate the new knowledge neurons. If a certain input is out of the editing scope, it fails to match
any memory, all inserted neurons will be inhibited after the activation function as shown in Figure 1.

In practice, edit input $x_i$ is shown in the form of a sequence of tokens such as "{*the*, *capital*, *of*,
*China*, *is*}" and $y_i$ is the single-token answer "*Beijing*". This indicates that we have a sequence of
hidden states $\{\mathbf{h}_1, \mathbf{h}_2, ..., \mathbf{h}_s\}$ corresponding to input $x_i = \{w_1, w_2, ..., w_s\}$. To avoid damaging the
original behavior of the edit model, the edit block merely works on the final token, which is the last
token before generation:

$$\text{ADAPTOR}(\mathbf{i}_j; \hat{\mathbf{K}}, \hat{\mathbf{V}}) = \begin{cases} 0 & j \neq s \\ \alpha \hat{\mathbf{V}}^\intercal g_{act}(\hat{\mathbf{K}}^\intercal \mathbf{i}_j) & j = s \end{cases}. \tag{7}$$

where $\mathbf{i}_j$ is the input corresponding to the $j$-th hidden state $\mathbf{h}_j$ in the sequence. As a result, only
when the entire input sequence is fed into the model, the new knowledge is activated, which not
only prevents the dramatic change to the original model but also benefits the gradient update to the
key-value pairs[2].

**Fine-tuning adaptors with multiple objectives**. While the above initialization effectively builds the
mapping from a certain edit input to the edit output, its impact on irrelevant knowledge may lead to

---

[1]See Appendix A.1 for detailed description of initialization of $\hat{\mathbf{k}}^0$ and $\hat{\mathbf{v}}^0$.

[2]See the discussion of gradient back-propagation of $\hat{\mathbf{k}}$ and $\hat{\mathbf{v}}$ in Appendix A.2.

catastrophic forgetting [24] issue, which is caused by the extending key-value pairs of the adaptor. In other words, we expect $\text{ADAPTOR}(\mathbf{i}; \hat{\mathbf{K}}, \hat{\mathbf{V}})$ could dominate the output for each $x_i \in \mathcal{D}_{in}$ but maintain unchanged prediction for $x_i \in \mathcal{D}_{out}$ and $x_{<i} \in \mathcal{D}_{in}$. Inspired by the elastic weight consolidation for neural networks [25], we set optimization goals to retrofit $\Phi^*$ with the consideration of the following perspectives.

(1) To maximize the prediction of $y_i$ from the last layer, we maximize the probability of the ground truth edit output given the edit input:

$$\mathcal{L}_{edit} = -\log[\mathbb{P}_{f_\Phi^*}(y_i|x_i)] \tag{8}$$

(2) Even though $\mathcal{L}_{edit}$ enables models to fit the mapping from $x_i$ to $y_i$ effectively, it may push our adaptor far from the initialization, which may damage the initialized key distribution and lead to overfitting. Hence, we propose an additional term to prevent the dramatic change of the update of $\hat{\mathbf{k}}$:

$$\mathcal{L}_{rec} = ||(\hat{\mathbf{k}}^0 - \hat{\mathbf{k}})^\intercal \mathbf{i}||_2^2 \tag{9}$$

(3) Importantly, to prevent the fine-tuning from changing the irrelevant knowledge, we sample some out-of-scope edit data to form $\mathcal{D}_{out}$[3] and retain the original outputs from the model:

$$\mathcal{L}_{irr} = -\frac{1}{|\mathcal{D}_{out}|} \sum_{(x_i,y_i)\in\mathcal{D}_{out}} \max(\hat{\mathbf{k}}^\intercal x_i - \theta, 0) \tag{10}$$

Hence, we comprehend each aspect to form the final objective to retrofit the key-value adaptor:

$$\mathcal{L} = \mathcal{L}_{edit} + a\mathcal{L}_{rec} + b\mathcal{L}_{irr} \tag{11}$$

where $a, b$ are hyper-parameters denoting the importance of the different objective aspects. It is worth noting that we edit one knowledge neuron at one time, but we still support sequential editing by iteratively inserting key-value pairs. During training, all parameters except for $\hat{\mathbf{k}}$ and $\hat{\mathbf{v}}$ for the current edit are frozen. That is, we freeze the prior edit knowledge neurons and simply update the neuron inserted for current edit. This procedure repeats until we have conducted edit over the entire dataset. Compared with parameter high-efficient tuning methods [19, 26], which injects the new knowledge into a pre-trained LM as a whole, iReVa focuses on editing parameters in a traceable manner. In other words, we can locate the edited knowledge neurons. At the end, we display the training procedure of iReVa in Algorithm 1.

---

**Algorithm 1** Training Procedure of iReVa

1: **Input** In-scope editing pairs $\mathcal{D}_{in}$; out-of-scope editing pairs $\mathcal{D}_{out}$; Original model $f_\Phi$; Iteration number $T$
2: **Initial** $\Phi^* \leftarrow \Phi$
3: **for** $(x_i, y_i) \in \mathcal{D}_{in}$ **do**
4:     **Initial** $\hat{\mathbf{k}} \leftarrow \mathbf{i}$; $\hat{\mathbf{v}} \leftarrow \mathbf{W}_{[y_i,:]}$         $\triangleright$ Initialize key-value adaptor as shown in Section 4.1
5:     $\Phi^* \leftarrow \Phi^* \bigcup \hat{\mathbf{k}} \bigcup \hat{\mathbf{v}}$
6:     **for** $t = \{1, 2, .., T\}$ **do**
7:         $\mathcal{L} \leftarrow \mathcal{L}_{edit} + a\mathcal{L}_{recon} + b\mathcal{L}_{irr}$     $\triangleright$ Retrofit key-value adaptor as shown in Section 4.2
8:         $\hat{\mathbf{k}} \leftarrow \text{Adam}(\hat{\mathbf{k}}, \nabla_\mathcal{L} \hat{\mathbf{k}})$
9:         $\hat{\mathbf{v}} \leftarrow \text{Adam}(\hat{\mathbf{v}}, \nabla_\mathcal{L} \hat{\mathbf{v}})$
    **return** $f_{\Phi^*}$

---

### 4.3 Activate Max-Matching Key in Adaptor (Inference Phase)

As we iteratively append $\hat{\mathbf{k}}$ and $\hat{\mathbf{v}}$ to the knowledge neurons. The above procedure will sequentially generate mappings from the edit input to the edit output. Eventually, we obtain two concatenated matrices $\hat{\mathbf{K}} \in \mathbb{R}^{d_1 \times n}$ and $\hat{\mathbf{V}} \in \mathbb{V}^{n \times d_1}$. During inference, we further control the amount of active neurons and highlight the max-matching memory. To this end, we introduce a max-pooling layer to extract the memory with the maximum matching score:

$$\text{ADAPTOR}(\mathbf{i}; \hat{\mathbf{K}}, \hat{\mathbf{V}}) = \alpha \hat{\mathbf{V}}_j^\intercal g_{act}(\hat{\mathbf{K}}_j^\intercal \mathbf{i}), \tag{12}$$

---

[3]Here, $\mathcal{D}_{out}$ is generated randomly. See Appendix A.4 for details.

where $j = \text{argmax}_t(\hat{\mathbf{K}}_t^{\mathsf{T}} \mathbf{i})$ and $\hat{\mathbf{K}}_t$ denotes the $j$-th column of $\hat{\mathbf{K}}$. As we can see, when there comes a new input, this layer will highlight the inserted knowledge neurons with the highest similarity score to the input as shown in Figure 1. It is worth noting that we exclude the max-pooling layer during the training procedure because this may impede the back-propagation due to the inactivation of the neurons.

# 5 Experimental Setup

## 5.1 Datasets

We perform extensive experiments on two modeling editing tasks: **zsRE** [8] is a commonly used model editing tasks derived from question-answering benchmark. Totally $19,086$ examples are included, each example includes a source question, paraphrase question and corresponding answer. We construct another **PARAREL** [27] dataset. Each sentence in PARAREL is derived from a triplet $(s, r, o)$ and the object $o$ was replaced with a "*[MASK]*" token and a paraphrased version is also involved. To apply PARAREL in model editing task, we selected those sentences that end with "*[MASK]*" token to conform to the format of next-token-prediction[4]. For both datasets, we sample irrelevant question-answer pair from NQ to evaluate the preservation to out-of-scope editing. We test 10K edit in a batch and denote them as **zsRE-**10**K** and **PARAREL-**10**K**, respectively.

## 5.2 Baselines

We compare our iReVa with 6 advanced baselines that support batch editing: **NO EDITING** denotes we do not modify the base model and utilize its original prediction; **FT** [28] is the simple fine-tuning with a constraint on the key-value adaptor. **MEMIT** [10] and **ROME** [9] are two methods employing a casual analysis to detect the most significant hidden states. They view the editing as a minimum optimization and edit the weight directly, which is effective in batch edit; **MEND** [8] applies rank-one decomposition to divide the model into two rank-one matrices, which is able to carry mass knowledge in the dense metrics; **MELO** [12] activates specific LoRA block corresponding to specific queries for multiple edits, which support large-scale editing in just one process.

## 5.3 Evaluation Metrics

We follow the commonly-used evaluation metrics [9, 10] to measure the effect of our editing method.

1. **Edit Success** (ES) measures the models' prediction accuracy on edited data $x_i \in \mathcal{D}_{in}$ by calculating $ES = \frac{1}{N} \sum_{i=0}^{N} \mathbb{I}(y_i = f_\Phi(x_i))$, which represents whether the new knowledge is successfully injected into the base model.
2. **Generalization** (Paraphrase Success, PS) measures the models' prediction accuracy on paraphrase questions provided by benchmarks. We compute paraphrase success with the same formulation but for $x_i$ in paraphrase questions set. Paraphrase success indicates whether the model can recognize similar expressions and provide edited answers.
3. **Specificity** (Neighborhood Success, NS) measures the models' prediction accuracy on irrelevant questions. Different from $\mathcal{D}_{out}$, these questions are only used for preventing data leakage. We compute neighborhood success with the same formulation but for $x_i$ in neighborhood questions set. Neighborhood success manifests the capability of solving catastrophic forgetting and preserving irrelevant knowledge stored in model.
4. **Score** is the average of three aforementioned metrics.

## 5.4 Implementation Details

Regarding editing datasets, we pre-process the edit input-output pairs following existing studies [8]. If the multiple tokens form a single prediction, we decompose the multiple tokens into multiple data pairs by greedily appending the previous token in the edit output at the end of the edit input[5]. For model selection, we conduct the experiments on GPT2-XL (1.5 Billion parameters) [29] due to its wide application on existing model editing studies. We trained iReVa on a single NVIDIA

---

[4]Appendix A.6 demonstrates the pre-processing step to PARAREL in detail.

[5]The processing procedure is displayed in Appendix A.5

A800 80G GPU. On two evaluated benchmarks, we set $a = 1e - 3, b = 1e - 3, \alpha = 2e - 1$, and iReVa is applied in 47-th (48 layers totally) layer inspired by the assertion in [15]. For the margin in activation function, we set $\theta = 0.75$ for zsRE, $\theta = 0.65$ for PARAREL. During training, we conduct experiments on `GPT2-XL` with setting learning rate as $5e - 2$, batch size as 1, epoch number as 5. We set the learning rate as $5e - 3$ for `GPT-NEO-2.7B`. More implementation details of baselines is displayed in Appendix A.7. We re-implement the comparable baselines using the same configuration reported in existing studies.

## 6 Results and Analyses

### 6.1 Comparisons to Existing Methods

Table 6.1 exemplifies performances of iReVa and baselines on zsRE and PARAREL with 10K edits in batch. As we can see, iReVa outperforms all baselines on average scores with noticeable margins. Even without retrofitting, our method is able to outperform the SOTA results by around $9\%$ and $6\%$ average score improvement on zsRE-10K and PARAREL-10K, respectively. Among all the baseline methods, FT achieves good results on ES and PS, this indicates that fine-tuning is simple but effective to inject knowledge but it could easily distract the irrelevant knowledge, resulting in a poor NS. Whereas other baselines can not guarantee the editing success in a batch, resulting in poor ES and PS. In comparison, iReVa achieves impressive results on all the evaluation metrics. It achieves close to $100\%$ ES without detriment to the original NS. We observe a slight improvement from the results of iReVa to iReVa+$\mathcal{L}$ on zsRE-10K dataset, it verifies our rationale deduce for the initialization of key-value pairs. However, the improvement brought by fine-tuning is not maintained on PARAREL-10K, we suspect this is because the involvement of irrelevant knowledge brings in little unexpected noise with possibility.

Table 1: Editing results on various model editing tasks with `GPT2-XL` as the base model. In our methods, $+\mathcal{L}$ represents iReVa with fine-tuning as described in Section 4.2.

| Method | zsRE-10K | | | | PARAREL-10K | | | |
| --- | --- | --- | --- | --- | --- | --- | --- | --- |
| | Score | ES | PS | NS | Score | ES | PS | NS |
| NO EDITING | 24.17 | 22.89 | 21.96 | 27.65 | 20.03 | 18.66 | 17.24 | 24.18 |
| FT | 57.29 | 82.80 | 64.51 | 24.57 | 52.64 | 83.32 | 53.06 | 21.55 |
| MEND | 15.94 | 12.43 | 12.04 | 23.35 | 0.16 | 0.00 | 0.00 | 0.50 |
| ROME | 11.10 | 17.26 | 14.24 | 1.80 | 5.35 | 9.65 | 6.23 | 0.17 |
| MEMIT | 42.51 | 52.62 | 47.29 | 27.63 | 46.17 | 62.60 | 52.71 | 23.20 |
| MELO | 32.51 | 42.75 | 28.12 | **26.65** | 25.95 | 34.19 | 20.83 | 22.83 |
| iReVa | 66.27 | **97.88** | 74.89 | 26.03 | **58.17** | **93.49** | 56.86 | 24.18 |
| iReVa $+\mathcal{L}$ | **66.77** | 97.47 | **76.38** | 26.47 | 56.80 | 89.85 | 56.37 | **24.18** |

### 6.2 Edit Withdrawal Test

Compared with the existing editing methods, our method has the unique advantage of interpretability and traceability, that is we can clearly identify the edit for each newly inserted key-value pair. This provides a chance to conduct an edit withdrawal test. Specifically, we test, after editing on 10K examples, if iReVa is able to withdraw certain edits and recover the original output from the base model without much loss. To this end, we inhibit corresponding knowledge neurons as withdrawing the edit, which is denoted as $f_{\Phi^*}^{-\hat{\mathbf{k}}}$. For evaluation, we introduce two metrics, namely **Retrieve Success** and **Consistency**. They are formulated as $RS = \frac{1}{N} \sum_{i=0}^{N} \mathbb{I}(f_{\Phi^*}(x_i) \neq f_{\Phi^*}^{-\hat{\mathbf{k}}_i})$ and $Con = \frac{1}{N} \sum_{i=0}^{N} \mathbb{I}(f_{\Phi}(x_i) = f_{\Phi^*}^{-\hat{\mathbf{k}}_i})$, respectively. The evaluation result on zsRE-10K is shown in Table 6.2. The results which are close to $100\%$ proves that iReVa can explicitly manipulate the activation of knowledge neurons and easily withdraw the updated knowledge. It is worth noting that this test is not applicable to any other editing methods as their edited parameters are untraceable. This is the first attempt at conducting more flexible knowledge editing.

Table 2: Results of edit withdrawal on zsRE-10K dataset with `GPT2-XL` as the base model.

| Method | Retrieve success | Consistency |
|--------|------------------|-------------|
| iReVa  | 98.02%           | 93.03%      |

## 6.3 Efficiency Analysis

We discuss the spatial and time complexities of iReVa. Regarding time complexity during inference, iReVa only insert the adaptor in a single $l$-th layer and the insertion only affects the final token prediction of the input. With $\mathbf{i} \in \mathbb{R}^{1 \times d_1}, \hat{\mathbf{K}} \in \mathbb{R}^{d_1 \times n}, \hat{\mathbf{V}} \in \mathbb{R}^{n \times d_1}$, the extra time consumption is $\mathcal{O}(d_1^2 n)$, which is unrelated to the input length and number of layers. Regarding spacial complexity, as we insert two vectors for each edit in a single layer, the extra spacial consumption is $\mathcal{O}(2nd_1)$. In practice, for `GPT2-XL` with 1.5B parameters, the adaptor merely possesses 0.08B parameters with 10K edits. There is no additional spacial complexity involved in the training phase, given that only $2d_1$ parameters are learnable for each edit. We empirically record that 10K edits with iReVa cost 7.5/1.6 hours (fine-tuning/without fine-tuning) with a single NVIDIA A800 GPU, compared to 9.16 hours for ROME and 5.4 hours for MEMIT.

## 6.4 Ablation Study

Table 6.4 shows iReVa's performance on zsRE-10K when we iteratively remove sub-modules: (1) w/o activation function denotes that we remove the activation function proposed in Equation 6. (2) w/o max-pooling denotes that we involve all knowledge neurons during inference instead of the design of Equation 12. (3) w/o $\mathcal{L}_{rec}$ denotes that we train iReVa without initialization and set $a = 0$ in Equation 11. (4) w/o $\mathcal{L}_{irr}$ means we do not apply $\mathcal{L}_{irr}$ by setting $b = 0$ in Equation 11. As we can see, all the modules contribute to the good results. In comparison, the activation function is important to preserve the out-of-scope edit. Without activation function, we can attain better results on ES and PS, but NS will decrease sharply. We also find that the influence of max-pooling is significant, which may attribute to noisy data added by a large amount of active but irrelevant knowledge neurons. Besides, excluding $\mathcal{L}_{rec}$ will lead to an observable drop on the three metrics because we discord the effective initialization on $\hat{\mathbf{K}}$ and $\hat{\mathbf{V}}$. Finally, disabling $\mathcal{L}_{irr}$ may induce a marginal improvement in ES and PS, but at the cost of a reduction in NS.

Table 3: Results of ablation study on zsRE dataset with `GPT2-XL` as the base model.

| Activation function | Max pooling | Loss $\mathcal{L}_{rec}$ | Loss $\mathcal{L}_{irr}$ | Score | ES | PS | NS |
|---------------------|-------------|--------------------------|--------------------------|-------|-------|-------|-------|
| ✓ | ✓ | ✓ | ✓ | 66.77 | 97.47 | 76.38 | 26.47 |
| ✓ | ✓ | ✓ | ✗ | 67.00 | 97.84 | 76.73 | 26.43 |
| ✓ | ✓ | ✗ | ✓ | 63.22 | 92.28 | 73.25 | 24.13 |
| ✓ | ✗ | ✓ | ✓ | 44.93 | 56.07 | 52.41 | 26.31 |
| ✗ | ✓ | ✓ | ✓ | 60.27 | 99.41 | 78.52 | 2.87 |

## 6.5 Generalization Capabilities of iReVa

**Layer generalization**. To evaluate the effect of iReVa in various layers, we iteratively apply iReVa and the other two baseline editing methods to different layers of `GPT2-XL`, which consists of 48 layers in total. Figure 6.5 illustrates the influence of three metrics on different layers with intervals. The tendency shows that the edit in the higher layer results in better editing results. This indicates that LMs' final prediction primarily depends on the information retrieved from higher layers and the knowledge stored in lower layers may be overshadowed. For ROME and MEMIT, apparently, they show distinct generalizations in edit layer. Their ES and PS peak at middle layer like 17 or 22, which proves that the layer generalization is remarkably relevant to the characteristics of different methods. Even though MEMIT achieves good performance in NS when the edit happens in lower layers, overall iReVa outperforms the baselines regarding the comprehensive evaluation metrics.

**LMs generalization**. We also test iReVa on different LLMs as base models, Figure 6.5 shows iReVa's generality on different backbones. We apply a larger LM `GPT-NEO-2.7B` [30] and smaller LM `GPT2-LARGE` [29] to evaluate the effect of iReVa on LMs with different sizes. Both

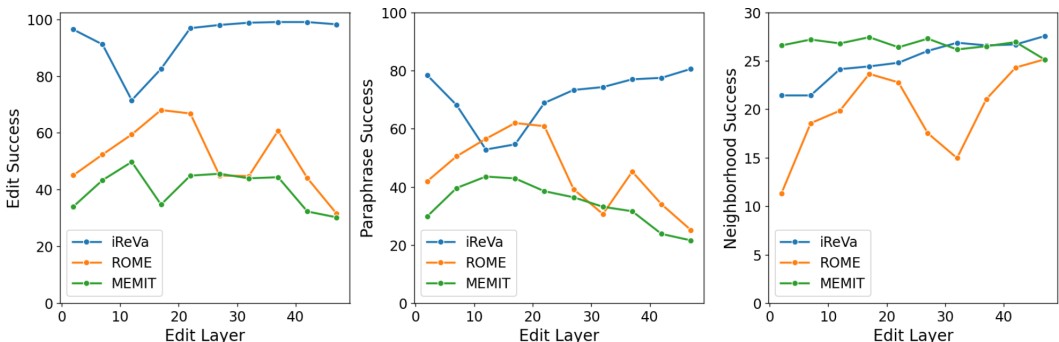

Figure 2: Results of edits in various layers on zsRE dataset with `GPT2-XL` as the base model.

Table 4: Results on zsRE dataset with `GPT2-LARGE`, `GPT-NEO-2.7B` as the base models.

| Engine | Method | Score | ES | PS | NS |
|---|---|---|---|---|---|
| `GPT2-LARGE` | ROME | 29.09 | 38.59 | 36.41 | 12.27 |
| | MEMIT | 43.72 | 56.25 | 49.25 | 25.67 |
| | iReVa | 62.41 | 91.22 | 72.36 | 23.65 |
| `GPT-NEO-2.7B` | ROME | 34.56 | 49.43 | 45.61 | 8.64 |
| | MEMIT | 59.68 | 80.83 | 69.38 | 28.83 |
| | iReVa | 62.20 | 88.23 | 70.71 | 27.66 |

`GPT-NEO-2.7B` and `GPT-LARGE` contain two-layer-FFN MLP blocks. IReVa can be deemed as a plug-in module for general LMs, which can be applied to more LMs. From the figure, we observe that iReVa can achieve the best average score on both LMs, which shows its general effect.

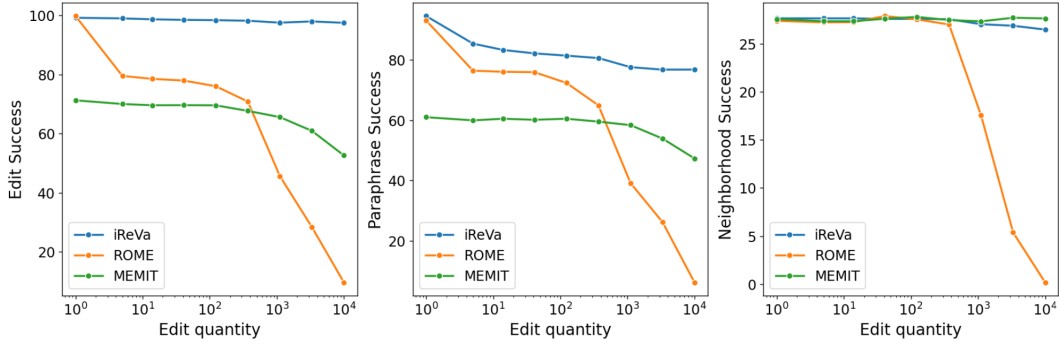

Figure 3: Results of edits with various size on zsRE dataset with `GPT2-XL` as the base model.

**Edit quantity generalization**. We discuss the influence on iReVa's performance with the variation of edit quantity, we simply increase the number of edits in the batch and evaluate ES, PS and NS. Figure 6.5 shows the tendency of three metrics along with the comparison to baselines ROME and MEMIT. As we can see, iReVa is robust to the number of edit in the batch. It consistently surpasses the other baselines when dealing with the various number of edits. MEMIT performs poorly even with a small number of edits. ROME drops dramatically as the edit number grows.

# 7 Conclusions

In this paper, we propose iReVa, a model editing method with traceable knowledge storage, which inserts edit key-value adaptor into the MLP module of a transformer model explicitly. iReVa displays prominent abilities of edit success, generalization and specificity and outperforms baselines with an observable margin. Besides, iReVa first successfully demonstrates its capacity on the knowledge withdrawal. For further research, we will focus on generalize iReVa to more LM architectures.

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

# A Appendix

## A.1 Detailed Description of Initialization of Key-Value Adaptor

We describe how we initialize $\mathbf{k}$ and $\mathbf{v}$ in detail. Given the input $x_i = \{w_1, w_2, ..., w_s\}$, we first obtain the corresponding embeddings for each token, such that $\mathbf{x}_i = \{\mathbf{w}_1, \mathbf{w}_2, ..., \mathbf{w}_s\}$. After encoded via $l$ Transformer layers, we obtain a sequence of hidden representations as input $\{\mathbf{h}_1^l, \mathbf{h}_2^l, ..., \mathbf{h}_s^l\}$. In the two-layer-FFN MLP block of $l$-th layer, after self-attention and layer norm, we have the hidden representation of the last token as:

$$\mathbf{i}_s^l = \text{LAYER\_NORM}(\mathbf{h}_s^l + \text{SELF\_ATTN}(\mathbf{h}_s^l))$$
$$\mathbf{o}_s^l = \mathbf{V}^{l\intercal} g_{act}(\mathbf{K}^{l\intercal}\mathbf{i}_s^l)$$
$$\mathbf{h}_s^{l+1} = \text{SELF\_ATTN}(\mathbf{i}_s^l + \mathbf{o}_s^l)$$

We extract $\mathbf{i}_s^{l+1}$ as the initialization of $\hat{\mathbf{k}}^0$. Subsequently, $\{\mathbf{h}_1^{l+1}, \mathbf{h}_2^{l+1}, ..., \mathbf{h}_s^{l+1}\}$ are further processed via the higher layers. In the last layer, we make prediction based on the hidden representation in $L$-th layer, which can be denoted as:

$$P_{f_\Phi}(y_i|x_i) = \text{SOFTMAX}(\mathbf{W}^\intercal \mathbf{h}_s^L),$$

where $\mathbf{W} \in \mathbb{R}^{d_1 \times |V|}$ and each column denotes the representation of a token. We extract the column corresponding to the ground truth edit out token $y_i$, that is $\hat{\mathbf{v}}^0 = \mathbf{W}_{[:,y_i]}$.

## A.2 Discussion of Back Propagation of Key-Value Adaptor

398 Recall the knowledge neurons of our key-value adaptor are:

$$\mathbf{o} = \mathbf{v}^\intercal g_{act}(\mathbf{k}^\intercal \mathbf{i}) + \hat{\mathbf{v}}^\intercal g_{act}(\hat{\mathbf{k}}^\intercal \mathbf{i})$$

399 Given $\mathcal{L}$, the gradients are computed as:

$$\frac{d\mathcal{L}}{d\hat{\mathbf{k}}} = g'_{act}(\hat{\mathbf{k}}^\intercal \mathbf{i}) \cdot \hat{\mathbf{v}} \cdot \hat{\mathbf{i}}^\intercal \frac{d\mathcal{L}}{d\mathbf{o}}$$

$$\frac{d\mathcal{L}}{d\hat{\mathbf{v}}} = g_{act}(\hat{\mathbf{k}}^\intercal \mathbf{i}) \frac{d\mathcal{L}}{d\mathbf{o}}$$

$$\frac{d\mathcal{L}}{d\mathbf{i}} = [g'_{act}(\mathbf{k}^\intercal \mathbf{i})\mathbf{v}^\intercal \mathbf{k} + g'_{act}(\hat{\mathbf{k}}^\intercal \mathbf{i})\hat{\mathbf{v}}^\intercal \hat{\mathbf{k}}] \frac{d\mathcal{L}}{d\mathbf{o}}.$$

400 where $g'_{act}$ is the derivative of the activation function. We have multiple observations of the gradients:
401 First, we would like the newly inserted neuron to be activated initially, namely $g_{act} > 0$. Otherwise,
402 the gradients are close to $0$ and the neurons are likely to be dead. This is the reason why we initialize
403 the $\hat{\mathbf{k}}$ and $\hat{\mathbf{v}}$ with the consideration of having a high matching value of $\mathbf{k}^\intercal \mathbf{i}$. Second, when we update
404 $\hat{\mathbf{k}}$ and $\hat{\mathbf{v}}$, they are unrelated to $\mathbf{k}$ and $\mathbf{v}$, which makes it possible to isolate the irrelevant knowledge.

405 For the knowledge neurons without our key-value adaptor, we have the propagation:

$$\mathbf{o} = \mathbf{v}^\intercal g_{act}(\mathbf{k}^\intercal \mathbf{i}).$$

406 The gradients of $\mathbf{i}$ are computed as:

$$\frac{d\mathcal{L}}{d\mathbf{i}} = g'_{act}(\mathbf{k}^\intercal \mathbf{i})\mathbf{v}^\intercal \mathbf{k} \frac{d\mathcal{L}}{d\mathbf{o}}.$$

407 As we can see, excluding the key-value adaptor in the neuron makes the gradients simply derived
408 from $\mathbf{k}$ and $\mathbf{v}$, which maintains the original knowledge in the neurons.

## A.3 Influence of $\theta$ and $a$

410 The influence of $\theta$ is illustrated in A.3. The figure shows the trade-off between the three metrics
411 smoothly. The primary affected metric is **Neighborhood Success**, and **Edit Success** and **Paraphrased**
412 **Success** exhibit a slight downward trend. For $a$, we find that merely **Paraphrase Success** peaks
413 while $a = 1e - 2$, meanwhile **Edit Success** and **Neighborhood Success** do not continue to improve
414 with the increase of $a$.

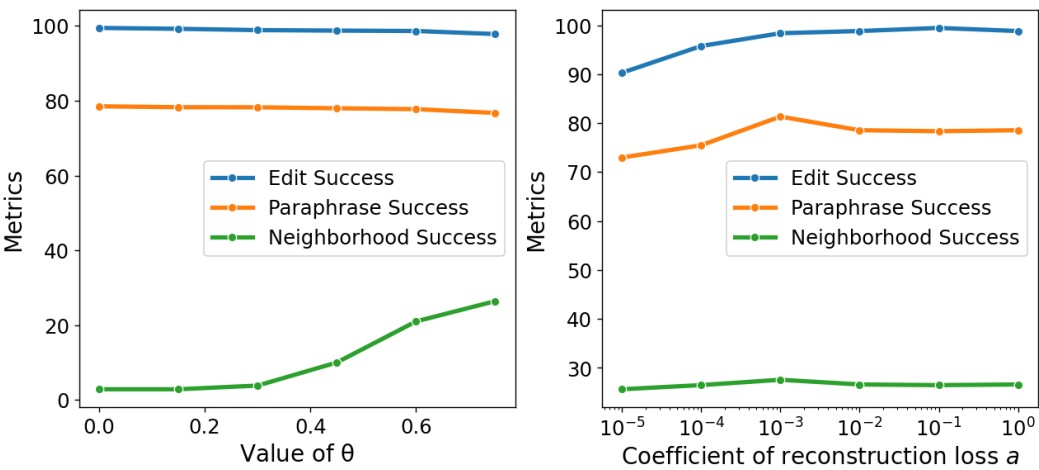

Figure 4: Correlation between three metrics and $\theta$(left) or $a$(right) of iReVa, ROME, MEMIT

## A.4 Sample off-scope examples for iReVa

To enhance iReVa's Specificity, we generate 3 kinds of irrelevant questions $q$ for each $(x, y) \in \mathcal{D}_{in}$ to minimize $\hat{\mathbf{K}}_i^{\mathsf{T}} \cdot x_{out}$, where $x_{out}$ is the representations of $q$. These questions are listed as follows: a) Randomly generated questions produced by feeding base model with a `bos` (begin of sentence) token. b) Questions generated by base model with feeding the subject $s$ of the $x$ provided by the benchmark. c) Questions sampled from other examples in training dataset, whose opinion is similar to contrastive learning [31]. During iReVa training, we generate 2 questions in a), 2 questions in b) and 6 questions in c) for each training example.

## A.5 Pre-processing procedure of zsRE

Shown in 2, we split each $(x, y)$ pair into multiple $(x', y')$ to ensure $y'$ is a single-token edit out. This procedure is also applied in the evaluation of zsRE and PARAREL, which measures the $(i+1)$-th token of edit-out prediction accuracy given edit-in and $i$ prefixes of edit-out.

---

**Algorithm 2** Pre-processing Procedure of PARAREL

1: **Input** Raw dataset zsRE $\mathcal{D}$, tokenization function encode;
2: **Init** $\mathcal{D}' = []$;
3: **for** $(x, y) \in \mathcal{D}$ **do**
4:     **Init** tokens = encode($y$);
5:     **for** $i \in \{0, 1, 2...\text{len}(\text{tokens}) - 1\}$ **do**
6:         $\mathcal{D}'.append((x + \text{tokens}[: i], y[i]))$;
    **return** $\mathcal{D}'$

---

## A.6 Pre-processing Procedure of PARAREL

---

**Algorithm 3** Pre-processing Procedure of PARAREL

1: **Input** Raw dataset PARAREL $\mathcal{D}$; Raw NQ dataset $\mathcal{D}_{loc}$; Function lcs computes the longest common sub-array of two strings, tokenization function encode, detokenization function decode;
2: **Init** $\mathcal{D}' = []$;
3: **for** $(r_i, v_i) \in \mathcal{D}$ **do**           ▷ For each relation and in-relation questions in $\mathcal{D}$
4:     **for** $(b_{ij}, a_{ij}) \in v_i$ **do**     ▷ For specific questions, rephrased versions and answers in $v_i$
5:         **If** $\text{len}(b_{ij}) \le 1$, **then continue**;
6:         **Init** subject = $b_{ij}[0]$;
7:         **Init** compatible_questions = [];
8:         **for** $q_{ijk} \in b_{ij}[1 :]$ **do**
9:             subject = lcs(encode($q_{ijk}$), encode(subject));
10:             **If** $q_{ijk}.endswith(\text{"}[MASK]\text{"})$, **then** compatible_questions.$append(q_{ijk})$;
11:         src_question = compatible_questions[0];
12:         subject = decode(subject)
13:         **If** (subject = "") $\vee$ (subject = src_question), **then continue**
14:         rephrased_question = $random.choice$(compatible_questions[1 :]);
15:         $\mathcal{D}'.append((\text{src\_question}, a_{ij}, \text{rephrased\_question}, \text{subjcet}, \mathcal{D}_{loc}.next()))$
16: **return** $\mathcal{D}'$

---

This section details the pre-process method on close text dataset PARAREL [27]. PARAREL contains 34 types of relations $r$, with an average of 900 question bags $b$ per relation, totaling 27,738 distinct questions $q$. And for each question bag, around 9 rephrased versions are recorded with a sole answer $a$.

The entire pre-process algorithm is shown in 3. To make PARAREL applicable for next-token-prediction task, we reserve the sentences that end with special token "*[MASK]*". After a round of filtering, we removed question bags $b$ with only 1 valid sentence that ends with "*[MASK]*" for both **Edit Success** and **Paraphrase Success** need to be computed. During this filtering, we collect the subject of question $s$ bag by calculating the longest common sub-array of all $q \in b$ tokenized by `GPT2Tokenizer` [29] simultaneously for specific methods require the subject of a question. The

next screening occurs at $b$ whose subject $s$ is an empty string or identical to $b[0]$. With residual question bags $b'$, we choose $b'[0]$ as the source question and a randomly sampled question from $b'[1:]$ as the paraphrase question.

Empirically, we believe PARAREL is harder than zsRE because the average token length of edit target is shorter, thus model can't give more empirical predictions based on given prefix of the target, which is mentioned in A.5. In other word, the account for first-token prediction may influence the difficulty of datasets noticeably.

## A.7 Implementation Details of Comparable Baselines

### A.7.1 Fine Tuning(FT)

We implement fine tuning on two feed-forward networks(`mlp.c_fc, mlp.c_proj`) at the layer of 46 with `GPT2-XL`. Base model is trained for 20 epochs with $lr = 1e-4$, batch size $= 32$.

### A.7.2 MEND

We do not load the pre-trained MEND [8] weight, but apply MEND directly. Hyper-parameters of MEND keep consistent with the configuration of MEND's open-source code.

### A.7.3 ROME, MEMIT

ROME [9] and MEMIT [10]'s setups on **GPT2-XL** also remain identical to the source code. On `GPT-NEO-2.7B`, we alter the edit layer to 5 for ROME and {3,4,5,6,7,8} for MEMIT.

### A.7.4 MELO

Due to larger edit amount and different backbone for zsRE, we modify several configurations to make MELO [12] comparable to our methods. For MELO's code book, we enlarge the number of blocks (clusters) to 100. Besides, we rewrite MELO's training loss to make it compatible with causal decoder.

