# OpenReview forum: "Initializing and Retrofitting Key-Value Adaptors for Traceable Model Editing"
_NeurIPS.cc/2024/Conference — Submitted to NeurIPS 2024_

### Official Review · Reviewer_T9Qb · 2024-07-12

**Soundness:** 3
**Presentation:** 2
**Contribution:** 3
**Rating:** 5
**Confidence:** 4

**Summary:**

The paper addresses the tracable sequential model editing challenge by plugging in additional model components to a transformer MLP blocks. The proposed approach adds additional model components for each edit, allowing for traceability for each edit.

**Strengths:**

•	The results indicate that it is a strong approach compared to relevant literature and its performance is relatively stable when scaling it to thousands of edits.
•	The approach allows for "separability of each edit" which in turn allows for additional operations such as edit updation or deletion as showcased in the Edit withdrawal experiment.
•	The edit withdrawal experiment is both unique and intriguing, as the concept of removing edits appears to be a novel area of exploration.

**Weaknesses:**

•	The overall approach does not appear to be novel, as it closely resembles T-Patcher. Both iReVa and T-Patcher rely on inserting neurons for editing and using neuron activations to control when to run the patch/adopter. Furthermore, analysis of the editing approach across different layers reveals the same pattern as discussed in the T-Patcher paper which involves adding additional model components in the final layer for optimal results.

•	Experiments with T-Patcher are missing from the comparisons to the existing methods section. Given its similarity, T-Patcher should be included for comparison.

•	Although T-Patcher performs editing in batches, it still uses a single patch of neurons for each edit, making its editing similarly traceable. Thus, the paper's claim of a "stronger capacity for carrying traceable edits" seems unfounded.

•	The Edit Withdrawal Test section is hard to understand. How exactly was the experiment conducted? Were all edits removed or only a limited set? Detailed experimentations for this section are needed as it is the only use case of traceability explored in the paper.

•	Editing techniques that rely on code books with playback vectors e.g. GRACE would allow for edits to be removed. The authors should make it clear that the withdrawal test is not possible for the editing techniques that they have chosen for comparison.

**Questions:**

N/A

---

> ### Author Rebuttal · Authors · 2024-08-06
>
> ## Reply to reviewer T9Qb
> Thank you, Reviewer T9Qb, for your valuable feedback! We appreciate your recognition of our work. This response primarily addresses the questions you raised in your review.
>
> - Treat T-Patcher [1] as baseline (Weakness 2). The reason for missing of T-Patcher as baseline is specified in our global rebuttal, we hope our explain will address your concerns.
>
> - Stronger capacity for carrying traceable edits (Weakness 3). Regarding traceable edits, similar to T-Patcher, we construct an independent neuron for each piece of knowledge. These neurons explicitly map the question to the answer in the new knowledge. Additionally, these neurons are independent of each other (with batch_size=1 during training), as you noted in the Strength section. Concerning capacity, if you meant "volume", iReVa's space overhead for new knowledge insertion is 2nd, where n is the number of new knowledge (q & a pairs) and d is the size of the model's embedding size (hidden size). As the value of n increases, this overhead is slightly less than 2nd because some knowledge can be added through updates rather than inserts. Section 6.3 mentions that for 10K edits, the additional parameter size is 0.08B (1.5B for the base model), increasing the parameters by only 5% for 10K edits. The comparison with T-Patcher in global rebuttal may be relevant to this weakness, you can access that part for detail.
>
> - Withdrawal analysis (Weakness 4): We apologize for the unclear description of the withdrawal analysis. In iReVa, one piece of knowledge corresponds to one neuron. Therefore, in withdrawal analysis, we withdraw one neuron at a time (setting its corresponding Vi to a zero vector) and test whether the model's response to the knowledge associated with this neuron changes, potentially reverting to the unedited state. Batch rollback of knowledge neurons is also feasible. Due to the independence of the neurons, the effect of batch rollback is consistent with single rollback.
>
> - Code book-based method's withdrawal analysis (Weakness 5): GRACE [2] and MELO [3] use code books to manage batches of knowledge. However, in GRACE and MELO, each cluster center corresponds to a batch of knowledge. Therefore, to withdraw a single piece of inserted knowledge, one must withdraw all knowledge inserted in the same batch, affecting other knowledge that does not need to be withdrawn. If we reduce the batch size, the number of clusters will significantly increase, which will markedly affect the performance of these methods. In our revised version, we will also adjust the description of this part.
>
> - Reference:
>
> [1] Transformer-patcher: One mistake worth one neuron. In The Eleventh International Conference on Learning Representations, 2023.
>
> [2] Aging with grace: Lifelong model editing with discrete key-value adaptors. ArXiv, abs/2211.11031, 2022.
>
> [3] Melo: Enhancing model editing with neuron indexed dynamic lora, 2023.

---

> > ### Comment · Reviewer_T9Qb · 2024-08-13
> >
> > Thank you for your detailed response. While my concerns about T-Patcher have been addressed, I remain somewhat skeptical about its traceability compared to GRACE.
> >
> > The GRACE algorithm processes edits individually in a streaming fashion(batch size 1). Each edit typically initiates its own cluster, with the cluster radius being dynamically adjusted based on the labels of subsequent edits that fall within its scope. The default cluster radius is maintained at a relatively small value to minimize the probability of grouping unrelated edits. However, the possibility of unrelated edits being clustered together, while very low, cannot be entirely eliminated. The paper should have a detailed discussion on the traceability of edits in comparison to other works and a comparison between iReVa and T-Patcher as in the global rebuttal.
> > The default GRACE approach may impact performance. However, since iReVa does not share neurons across various edits, it could have performance trade-offs in comparison with T-Patcher.
> >
> > Given the current presentation of the work, I do not feel comfortable raising the score, I will be maintaining my current positive evaluation of your manuscript.

---

> > > ### Author Response · Authors · 2024-08-14
> > >
> > > Thank you for your insightful response! Your concern indeed requires further clarification, as there are some similarities between Method iReVa and Method GRACE when establishing adaptors for multiple edits. Therefore, we will further compare the traceability of iReVa and GRACE, and we hope this will help address your concern.
> > >
> > > - Similarity. In general, Method iReVa can be seen as a variant of Method GRACE, where each cluster is limited to only one edit, which is the cluster center itself. Additionally, the activation bias in iReVa can be considered similar to the cluster radius in GRACE. If, in a benchmark, it is rare for multiple edits to share the same edit target, then iReVa and GRACE are essentially similar because GRACE's codebook rarely involves the expand operation.
> > >
> > > - Difference. If, in a benchmark, multiple edits share the same edit target (e.g., country name), GRACE‘s codebook will experience a trade-off based on the size of the initiative radius. A radius that is too large can lead to errors and forgetting, while a radius that is too small can result in too many clusters. In contrast, Method iReVa always maintains one edit per neuron, avoiding the issue of edits forgetting. The phenomenon of forgetting often occurs during the split stage in the codebook. When the current edit has a similar input representation x to a certain cluster center C0 but has a different edit target y, GRACE's codebook will perform a split operation, creating a new cluster C1 from the current edit and splitting the radius between the new cluster and C0 based on their distance. In this process, the reduction in C0's radius can cause some edits that originally belonged to C0 to fall outside of C0. If these edits are fundamentally different from C0, the codebook loses these edits. You mentioned in your response that Method GRACE has a low probability of merging two unrelated edits into the same cluster. Considering that different clusters in GRACE's codebook do not overlap with each other, and the representation space for cluster centers is limited, when the number of edits is large (GRACE performs 1K edits whereas iReVa performs 10K edits), the increased number of clusters filling the space may involve more expand or split operations, potentially introducing more errors.
> > >
> > > Additionally, regarding the trade-off phenomenon you mentioned between Method iReVa, we discussed this in Section 4.3 of our paper. During training, although the neurons corresponding to previous edits are frozen during backpropagation, they are included in the forward propagation calculations. This mechanism will introduce noise to forward propagation so as to increase the robustness of the editing.
> > >
> > > These comparisons will be included in the revised paper after simplification. Thank you again for your valuable feedback.

---

### Official Review · Reviewer_QEHi · 2024-07-13

**Soundness:** 3
**Presentation:** 3
**Contribution:** 3
**Rating:** 7
**Confidence:** 3

**Summary:**

This paper introduces iReVa, a novel method for model editing that explicitly initializes and retrofits key-value pairs into MLP blocks of transformer models to perform CRUD (Create, Read, Update, Delete) operations on LMs. iReVa aims to update knowledge in LMs without damaging irrelevant knowledge, offering better interpretability and traceability compared to existing methods. The method is validated through experiments on GPT series models, showing significant improvements in edit success and generalization without affecting specificity.

**Strengths:**

Provision of the first attempt at conducting knowledge withdrawal tests for model editing methods.

The paper includes a comprehensive analysis of iReVa's performance, including knowledge withdrawal tests and generalization tests.

iReVa's approach to model editing is innovative, focusing on retrofitting key-value adaptors into MLP blocks for traceable model editing

**Weaknesses:**

This paper could benefit from a more detailed comparison with other model editing methods, especially those focusing on lifelong learning and continual editing [1][2].

It does not discuss the computational efficiency of iReVa in terms of inference time or memory, which is crucial for real-world applications.

The reliance on the hypothesis that factual knowledge is stored in MLP blocks may be limiting [3], and the authors could explore the broader implications of this assumption.

The method's applicability to other types of tasks, such as erasing hallucinations, is not validated.

There is a noticeable absence of experimental validation on other recent and updated models such as GPT-J (used by ROME etc.), LLaMA.

The technical novelty of iReVa is somewhat limited, as it builds upon existing concepts like MEMIT [4] and key-value memory structures in MLPs [2].

The absence of a strategy for selecting the adaptor layer may hinder the method's rapid migration and application to various language models。

Equation 3 requires clarification, why 'i' and 'o' in Equation 3 are both passed through SELF_ATTEN again?

References

[1] Aging with GRACE: Lifelong Model Editing with Discrete Key-Value Adaptors, Hartvigsen et al,
Neurips 2023.

[2] Transformer-Patcher: One Mistake worth One Neuron, Huang et al, ICLR 2023.

[3] What does the Knowledge Neuron Thesis Have to do with Knowledge? Niu et al, ICLR 2024

[4] Mass-Editing Memory in a Transformer, Meng et al, ICLR 2023.

**Questions:**

See weaknesses.

---

> ### Author Rebuttal · Authors · 2024-08-06
>
> ## Reply to reviewer QEHi
> Thank you, Reviewer QEHi, for your valuable feedback! We appreciate your recognition of our work. This response primarily addresses the questions you raised in your review.
>
> - More baselines (Weakness line 1): Indeed, the baselines you mentioned are worth comparing. We explain the reason for this in our global rebuttal and hope our specification will address your concern.
>
> - Inference computational efficiency (Weakness line 2): In Section 6.3, we mentioned that iReVa's theoretical inference time overhead compared to the base model is 2nd, where n is the number of new knowledge (q & a pairs) and d is the size of the model's embedding size (hidden size). This overhead is independent of the prompt sequence length L and constitutes only a small portion of the entire model's forward propagation. The inference space overhead is 2nd, consistent with the parameter size introduced by iReVa.
>
> - Knowledge storage (Weakness line 3): Indeed, recent works have proposed storing knowledge in the self-attention module, such as PMET [1]. We proposed iReVa to offer an interpretable and manageable model editing approach. However, the self-attention module is inherently complex, and deconstructing it into an interpretable forward propagation method requires more effort. In our future work, we look forward to proposing interpretable forward propagation in self attention modules.
>
> - Other tasks (Weakness line 4): Regarding the hallucination task, we conducted the experiment following GRACE [2] and Melo [3]. The experimental results are presented in the table below, with the baselines' results selected from the Melo [3]. The results show that iReVa does not perform well on the hallucination task because the target prompts in this dataset are too long. iReVa creates a key-value pair (knowledge neuron) for each token in the target prompt and its prefix. Due to the excessive number of introduced neurons and the high similarity of the prefixes, the probability of errors significantly increases. Additionally, due to the mechanism of equation 7 (Section 4.2), iReVa will only affect the last token of the prompt which is used for next token prediction. However, the hallucination task requires calculating the perplexity (PPL) of the entire sentence, and iReVa only modifies the hidden states of the token at the last position. Since GPT series models are unidirectional autoregressive models, the tokens before this position cannot access this additional information introduced by iReVa, resulting in poor performance of this method on the hallucination task. An intuitive solution is to apply this mechanism to every token in the sentence instead of just the last one, which is just the strategy we applied in the supplymentary experiment below, but this would also introduce more noise, leading to suboptimal performance.
>
> | Backbone | Method | ES $\downarrow$ | ARR $\downarrow$ | Locality $\downarrow$ |
> | :------: | :----: | :----: | :-----: | :------: |
> |          |  ROME  | 103.82 |  14.02  |   30.28  |
> | gpt2-xl  | GRACE  |  7.14  |  10.00  |   15.84  |
> |          |  Melo  |  1.04  |   2.66  |   17.45  |
> |          | iReVa  | 376.69 | 2312.39 |   13.76  |
>
> - More backbones (Weakness line 5). We additionally conducted the experiments in Section 6.5 (table 4) on gpt-j-6b [4] as a supplement, and the results are shown in the table below. iReVa also outperforms baselines on gpt-j-6b. Furthermore, the hyperparameters of the baselines were kept consistent with their source code. We did not include LLaMA [5] family models due to the following considerations: LLaMA models have a different FFN structure compared to common causal decoder models, containing three linear layers: up_proj, down_proj, and gate_proj, unlike the K, V structure in GPT models, hence some baselines can't be reproduced on LLaMA family models.
>
> | Backbone | Method | S $\uparrow$  | ES $\uparrow$ | PS $\uparrow$ | NS $\uparrow$ |
> | :------: | :----: | :---: | :---: | :---: | :---: |
> |          |  ROME  | 40.86 | 53.81 | 49.89 | 18.87 |
> | gpt-j-6b | MEMIT  | 66.41 | 94.04 | 72.48 | 32.70 |
> |          | iReVa  | 69.70 | 99.71 | 77.10 | 32.27 |
>
> - Strategy for selecting the adaptor layer (Weakness line 7): The selection of layer is certainly a direction worth exploring. In our early experiments, we examined the patterns of words output by each layer after providing a prompt to the given model. For the GPT-2 XL model, we found that most prompts resulted in an output of a meaningless word, such as a space or newline, at the penultimate layer. Therefore, we hypothesized that there might be unused knowledge storage capacity in this layer, which led us to select the penultimate layer as the adaptor layer. In current editing works, the editing layer is usually a given hyperparameter, and the optimal value is found by modifying the layer. Thus, we followed these works and conducted layer normalization experiments (Section 6.5). The penultimate layer performed slightly better than other layers, which to some extent supports this hypothesis.
>
> - Problems of Equation 3 (Weakness line 8): In equation 3, i and o are passed into Equation 3 due to the "add and norm" (residual connection) architecture in widely-recognized transformer models.
>
> - Reference:
>
> [1] Transformer-patcher: One mistake worth one neuron. In The Eleventh International Conference on Learning Representations, 2023.
>
> [2] Aging with grace: Lifelong model editing with discrete key-value adaptors. ArXiv, abs/2211.11031, 2022.
>
> [3] Melo: Enhancing model editing with neuron indexed dynamic lora, 2023.
>
> [4] GPT-J-6B: A 6 Billion Parameter Autoregressive Language Model, https://github.com/kingoflolz/mesh-transformer-jax, 2021.
>
> [5] LLaMA: open and efficient foundation language models, 2023.

---

> > ### Comment · Reviewer_QEHi · 2024-08-13
> >
> > Thank you for your reply, my concern has been addressed. I have raised my score.

---

### Official Review · Reviewer_vMmP · 2024-07-13

**Soundness:** 3
**Presentation:** 2
**Contribution:** 3
**Rating:** 5
**Confidence:** 4

**Summary:**

This paper introduces a novel method called iReVa for knowledge editing. iReVa initializes and retrofits key-value pairs into MLP blocks to create a new mapping of knowledge without affecting related information. Compared to existing methods, iReVa offers better interpretability and a stronger ability to make traceable edits.

**Strengths:**

1. The proposed methods demonstrate great performance compared to other baselines under the batch editing scenarios.

**Weaknesses:**

1. The color in the figure is not obvious to discriminate between the original knowledge neurons and new knowledge neurons.
2. The computation of the proposed method is similar to T-Patcher, I'm curious about the difference between them. The proposed methods are designed to tackle the batch edit, but it seems it still needs to add one neuron for each example.

**Questions:**

1. Line 57-59 focus on self-attention, while the author selects MLP to further investigate, the logic here is a bit confusing. Is there something I missing?

---

> ### Author Rebuttal · Authors · 2024-08-06
>
> ## Reply to reviewer vMmP
> Thank you, Reviewer vMmP, for your valuable feedback! We appreciate your recognition of our work. This response primarily addresses the questions you raised in your review.
>
> - Figure issue (Weakness 1): Thanks for your reminder. We will revise the figure and adjust the image color to make it clearer in the revised version.
>
> - T-Patcher [1] issue (Weakness 2): Indeed, the forward propagation process of iReVa is similar to T-Patcher. The difference between iReVa and T-Patcher is clarified in our global rebuttal, we wish its content will address your issue.
>
> - Line 57-59 (Questions 1): We apologize for the writing error. The sentence in lines 57-58 should be rewritten as: "In contrast, PMET [2], through a cosine similarity analysis on hidden states experiment, posed viewpoints that the self-attention module can extract various types of knowledge."
>
> - Reference:
>
> [1] Transformer-patcher: One mistake worth one neuron. In The Eleventh International Conference on Learning Representations, 2023.
>
> [2] Pmet: Precise model editing in a transformer. In AAAI, 2024.

---

> > ### Comment · Reviewer_vMmP · 2024-08-12
> >
> > Thank you for your detailed response, and I will be maintaining my positive evaluation of your manuscript.

---

### Official Review · Reviewer_QtC8 · 2024-07-13

**Soundness:** 3
**Presentation:** 3
**Contribution:** 3
**Rating:** 7
**Confidence:** 4

**Summary:**

This paper focuses on model editing at a low cost. Evidence suggests that modules carrying knowledge in a Transformer module are primarily the MLP blocks. Therefore, the authors propose a method, namely iReVa, to initialize and retrofit key-value pairs into MLP blocks in a Transformer for explicitly inserting new knowledge. Specifically, they insert new neurons in the MLP blocks for each piece of knowledge. Each neuron is initialized with the embedded key and value derived from the input-output pair, respectively. To prevent dramatic change to the irrelevant knowledge, iReVa further retrofits the key and value by fine-tuning with multiple objectives. Compared to the existing methods such as MEND, ROME, MEMIT, and MELO, iReVa reveals better interpretability and stronger capacity for carrying traceable edits. The experiments on zsRE-10K and PARAREL-10K datasets reveal that iReVa has superior performance regarding edit success, generalization, and specificity. Further edit withdrawal test indicates that iReVa can explicitly manipulate the activation of neurons and easily withdraw the edits.

**Strengths:**

1.	This paper focuses on modeling editing, which has significant applications in the era of LLMs. It can be applied to alleviate the hallucination issue of LMs and resolve the out-of-date as well as missing knowledge in an LM.
2.	This paper introduces a novel editing method with key-value adaptors for traceable model editing. The proposed method makes sense to me. The initialization with embedded key and value derived from the input-output pair can easily make precise edits to the model. Further retrofitting refines the adaptors to satisfy the task.
3.	For experiments, the author has comprehensively shown the superiority of their method in the perspectives of edit success, generalization, and specificity. And more analyses reveal the generalization of iReVa. Particularly, the edit withdrawal test in Section 6.2 is well-designed, which shows the effect of traceable edits and could provide a potential solution for dynamic knowledge maintenance for LMs.
4.	Overall, this paper is well-written and easy to follow.

**Weaknesses:**

1.	The discussions on the limitations and broader societal impacts of iReVa are not included in the paper. I have some questions about the application scope of the proposed method. Please see the questions below.

Questions
1.	Could iReVa lead to a dramatically increasing number of parameters? Let’s see if there are millions of knowledge for editing, how can you potentially insert all the knowledge into LMs with iReVa?
2.	After you change a piece of knowledge, can the reasoning still be conducted for the edited knowledge? For example, if we have edited the president of America, could some reasoning questions like ``Who is the wife of the president of America” also be resolved with the new knowledge?
3.	Typo: ``evident’’ in line 6 should be ``evidence’’. Please check.

**Questions:**

Please see weaknesses.

**Limitations:**

No, the author should discuss the limitations of the proposed method such as the application scope, the potential risks, and future improvement to indicate how robust the results are to violate the
assumption. I would like the author to add such information during the rebuttal.

---

> ### Author Rebuttal · Authors · 2024-08-06
>
> ## Reply to reviewer QtC8
> Thank you, Reviewer QtC8, for your valuable feedback! We appreciate your recognition of our work. This response primarily addresses the questions you raised in your review.
>
> - Increasing the number of parameters (Question 1): In Section 6.3 of our paper, we analyzed the space overhead of iReVa, which is 2nd, where n is the number of new knowledge (q & a pairs) and d is the size of the model's embedding size (hidden size). This means that iReVa's space consumption grows linearly with the amount of new knowledge. Theoretically, iReVa’s space overhead could be smaller, but this would involve following techniques. In practice, a significant portion of the new knowledge added to the model are frequently-updated, meaning the answer changes while the question remains the same. For such knowledge, we only need to locate the corresponding neurons Ki, Vi. For example, when inserting new knowledge, we can directly feed the new question and supplymentary similar questions into the model with iReVa to see if any neurons are simultaneously activated by all questions (essentially a threshold-based similarity match). By modifying Vi to adapt to the new answer, no additional space is needed. For new facts, where the question has not been recorded by the model's memory via iReVa, additional space is required.
>
> - Reasoning for new knowledge (Question 2): Reasoning has always been a challenging issue in the field of model editing. In the background of large language models, reasoning ability lacks interpretability, making it difficult for most model editing methods, including iReVa, to apply newly learned knowledge in reasoning tasks. One existing attempt is IKE [1], which inspired from in-context learning. However, this approach affects interpretability and locality metrics. Overall, the reasoning ability of current model editing methods often trades off with interpretability, which many researchers are striving to solve.
>
> - Limitations: Thanks for raising this issue. The limitation of iReVa can be summarized as follows: 1. iReVa performs poorly when the target prompt is a long sentence because it constructs a knowledge neuron for each token in the target prompt, thereby increasing the training time cost. Additionally, during inference, the high number of neurons increases the probability of errors. 2. To maintain iReVa's interpretability, its application is limited, including that iReVa can be only applied on GPT like models and generation task. 3. The behaviour of iReVa (ES and PS) won't enhance noticeably as the scale of base model grows. These limitations will also be included in our revised version.
>
> - Reference:
>
> [1] Can we edit factual knowledge by in-context learning? Proceedings of the 2023 Conference on Empirical Methods in Natural Language Processing.
>
> [2] Editing factual knowledge in language models, 2021.
>
> [3] Fast model editing at scale, 2022.

---

> > ### Comment · Reviewer_QtC8 · 2024-08-13
> > **Reply for Authors' Response**
> >
> > Thanks for your reply. I have read your response carefully. I will keep my initial rating for your manuscript.

---

### Author Rebuttal · Authors · 2024-08-06

# Global rebuttal
We appreciate all reviewers' invaluable feedbacks! This section is our global rebuttal, which addresses common questions raised by multiple reviewers. We hope all reviewers will see this. The following is the content.

## Difference between iReVa and T-Patcher

- Mistaken knowledge: T-Patcher only establishes new neurons for knowledge where the model previously answered incorrectly, while iReVa establishes new neurons for all knowledge, which benefits subsequent neuron management. In real scenarios, the knowledge to be edited may need frequent updates, meaning it might be modified or rolled back in the future. iReVa's neuron management allows for deletion, updating, and other operations on identifiable neurons (by feeding new question and supplymentary similar questions into the model with iReVa to see if any neurons are simultaneously activated by all questions).

- Backbone difference: According to the experiment section of T-Patcher, they only apply T-Patcher on transformers of encoder-decoder architecture. T-Patcher's editing position is at the encoder's final layer, unlike iReVa, which operates at the penultimate layer. This will be explained in the following responses.

- Forward propagation differences: During inference, iReVa uses max pooling for newly inserted neurons to avoid unnecessary noise from activating too many neurons. Furthermore, we do not apply learnable bias for newly inserted vectors Ki and Vi.

- Initialization techniques: In iReVa, we initialize two parameters, Ki and Vi, for new neurons so that the model can answer questions related to new knowledge even without training, demonstrating iReVa's stronger interpretability.

- Different optimization goals: For trained iReVa, we do not directly optimize the activation value of qKi because iReVa operates at the penultimate layer. Optimizing qKi would only ensure correct output at this layer, not at the final layer. Operating iReVa at the final layer, as T-Patcher does, might result in loss of locality. However, operating at the penultimate layer allows for correction by the final layer's frozen parameters. Additionally, with initialized weights, we added constraints on learnable parameters to prevent them from deviating too far from the initial weights, as our experiments in Section 6.1 showed the effectiveness of our initialization.

- Independent neurons: iReVa establishes a dedicated neuron for each piece of new knowledge. In the algorithm 1 (Section 4.2, Line 167-170), We merely optimize one key-value pair (knowledge neuron) for current edit question, namely the parameter previous-edited question's key-value pairs is frozen, thus neurons for different knowledge are independent and do not interfere with each other. By re-feeding the question into the model, the corresponding neuron can be located. In T-Patcher, the batch size during training is set to 32, that means T-Patcher cannot distinguish which of these 32 neurons corresponds to which piece of knowledge.

## Missing of baselines

T-Patcher [1] and GRACE [2] both use encoder-decoder models as their backbone for generation tasks like zsRE [3], whereas iReVa uses GPT-like models. Specifically, these methods operate on the model's encoder. In the experiment of T-Patcher, only encoder only model and encoder-decoder model are used. If we force T-Patcher to embed in our GPT like backbone, the implementation will differ remarkably from their source code. For GRACE, they do not apply decoder-only models in zsRE, whereas they use GPT models on hallucination task, which means GRACE might be compatible with the GPT series models. Among the methods we compared, MELO [4] has a similar setup to GRACE and shows effectiveness, so we chose it as a representative for comparison.

- Reference:

[1] Transformer-patcher: One mistake worth one neuron. In The Eleventh International Conference on Learning Representations, 2023.

[2] Aging with grace: Lifelong model editing with discrete key-value adaptors. ArXiv, abs/2211.11031, 2022.

[3] Fast model editing at scale, 2022.

[4] Melo: Enhancing model editing with neuron indexed dynamic lora, 2023.

---

### Author Response · Authors · 2024-08-12
**Thanks for your constructive reviews**

Dear Reviewers,

We sincerely thank you for your meticulous and insightful feedback on our manuscript. Your valuable comments have significantly improved the quality of our work, and we greatly appreciate the time and dedication you devoted to reviewing our paper.

We wanted to kindly follow up and see if you’ve had a chance to review our response. We hope it has addressed the concerns you raised in your review. If there are any additional questions or if further clarification is needed, we would be more than happy to assist.

Thank you again for your valuable time and thoughtful consideration.

Best regards,
The Authors

---

### Decision · Program_Chairs · 2024-09-25

**Decision:**

Reject

**Comment:**

The paper presents a model editing method that explicitly targets MLP blocks by retrofits key-value pairs of new piece of knowledge. One of the major limitations of the submitted work is the lack of comparison with obvious baselines that are very similar to the method proposed in the paper. From the rebuttal, it is clear that authors were familiar with those work and they did a good job in explaining the differences. The discussion with the related work on whether an approach can be applied to a certain class of model or not should be done in the paper. Authors should clearly articulate the contribution of their work with respect to other methods in the paper. This is inline with the novelty concern raised by the reviewers and I did not see a response to that weakness.

Since the authors mention that the closest approach to their method, T-Patcher, can not be applied to decoder only model given the current implementation, does iReVa can be applied to encoder only models? If yes, it is natural to think of a comparison given that both approaches are very similar and it is unclear whether the various design decisions taken by iReVa as mentioned in the Global Rebuttal are the reason of better editing or not. If iReVa can not be applied to encoder models, this should be clearly stated.

The paper also lacks a limitation section which is not compulsory but it is important to clearly state the limitations of the work. The checklist provided by the authors is incomplete and did not provide response to any question which kills the sole purpose of the requirement of submitting the checklist.